# Prediction of minimal hepatic encephalopathy by using an radiomics nomogram in chronic hepatic schistosomiasis patients

Ying Li[1], Shuai Ju[2], Xin Li[1], Yan Li Zhou[3], Jin Wei Qiang[1]*

1 Department of Radiology, Jinshan Hospital, Fudan University, Shanghai, China, 2 Department of Intervention Radiology, Jinshan Hospital, Fudan University, Shanghai, China, 3 Department of Nuclear medicine, Jinshan Hospital, Fudan University, Shanghai, China

* dr.jinweiqiang@163.com

## Abstract

### Objective

To construct an MR-radiomics nomogram to predict minimal hepatic encephalopathy (MHE) in patients with chronic hepatic schistosomiasis (CHS).

### Methods

From July 2017 to July 2020, 236 CHS patients with non-HE (n = 140) and MHE (n = 96) were retrospective collected and randomly divided into training group and testing group. Radiomics features were extracted from substantia nigra-striatum system of a brain diffusion weighted images (DWI) and combined with clinical predictors to build a radiomics nomogram for predicting MHE in CHS patients. The ROC curve was used to evaluate the predicting performance in training group and testing group. The clinical decisive curve (CDC) was used to assess the clinical net benefit of using radiomics nomogram in predicting MHE.

### Results

Low seralbumin (P < 0.05), low platelet count (P < 0.05) and high plasma ammonia (P < 0.05) was the significant clinical predictors for MHE in CHS patients. The AUC, specificity and sensitivity of the radiomics nomogram were 0.89, 0.90 and 0.86 in the training group, and were 0.83, 0.85 and 0.75 in the training group. The CDC analysis showed clinical net benefits for the radiomics nomogram in predicting MHE.

### Conclusions

The radiomics nomogram combining DWI radiomics features and clinical predictors could be useful tool to predict MHE in CHS patients.

**Data Availability Statement:** All relevant data are within the manuscript and its Supporting Information files.

**Funding:** YL received funding from Jinshan Science and Technology Committee (No. 2021-3-01). JWQ received funding from Shanghai Municipal Health Commission (No. ZK2019B01). The funders had no role in study design, data collection and analysis, decision to publish, or preparation of the manuscript.

**Competing interests:** The authors have declared that no competing interests exist.

## Author summary

Minimal hepatic encephalopathy (MHE) is usually neglected clinically in chronic hepatic schistosomiasis (CHS) patients. The diffusion change in substantia nigra-striatum system of MHE patients has been reported. We hypothesized that the change could be better detected by DWI-based radiomics. A radiomics nomogram combining radiomics and clinical predictors of MHE was built to predict MHE in CHS patients. The results demonstrate that the radiomics nomogram would be useful for predicting MHE in CHS patients.

## Introduction

Hepatic encephalopathy (HE), developed secondary to portal hypertension, is a serious complication of chronic hepatic schistosomiasis japonicum (CHS) next only to upper gastrointestinal bleeding [1,2]. Neurological symptoms of HE include personality change, attention deficits, sleep rhythms alteration and mild cognitive impairment (minimal hepatic encephalopathy, MHE) progressing to stupor and coma (overt hepatic encephalopathy, OHE) [2–4]. MHE is usually neglected clinically in CHS patients because of lacking biochemical evidence of intrinsic liver disease [1,5]. As MHE progresses, OHE occurs. Therefore, early detection of MHE can reduce the risk of development of OHE, facilitating HE prevention [6].

The exact pathophysiological mechanism of HE is still unknown. Evidences suggest that cytotoxic brain edema (astrocyte swelling) and substantia nigra-striatum system dysfunction are implicated in the pathogenesis of HE [1,3,5]. As a non-invasive tool, magnetic resonance imaging (MRI) has widely used to investigate the localization and pathophysiological mechanisms of brain functions, such as cognition and sensory perception [1,5,7]. An elevated ADC values on diffusion weighted imaging (DWI), increased mean diffusivity on diffusion tensor imaging (DTI), increased cerebral blood perfusion on arterial spin labeling (ASL) and abnormal metabolism on MR spectroscopy in substantia nigra-striatum system of HE patients were reported [5,6,8,9]. DWI, reflecting the diffusion of water molecules, has been proved to be useful for diagnosing HE in cirrhosis [6].

Radiomics, a method of high-throughput quantitative information extraction from medical images, has attracted increasing attention in recent years [10,11]. DWI-based radiomics features has been considered as a biomarker and used for predicting treatment effect in sarcoma [12]. A combined analyses of the radiomics features and clinical risk factors produce a radiomics nomogram, which is becoming the most promising approach for individualized clinical management [13].

We hypothesized that change of the water molecules diffusion in substantia nigra-striatum system of CHS patients could be detected by the DWI-based radiomics. The aim of this retrospective study was to develop a DWI-based radiomics nomogram for predicting MHE in CHS patients.

## Materials and methods

### Ethics statement

This retrospective study was approved by the Institutional Review Board of Jinshan Hospital, Fudan University and the requirement for informed consent was waived.

## Study design and participants

From July 2017 to July 2020, 355 consecutive CHS patients with brain MRI scanning were reviewed. The inclusion criteria were: (1) a history of schistosomiasis, (2) typical liver CT or ultrasonic findings of CHS, (3) patients admitted brain MRI scanning within 3 months before or after a diagnosis of MHE. The exclusion criteria were: (1) clinical diagnosed OHE (n = 9); (2) history of using any drugs with liver or central nervous system toxicity (n = 2); (3) patients with an upper gastrointestinal bleeding or serious infection recently (n = 6); (4) lacking of DWI or having obvious artifacts on MRI (n = 2).

CHS was diagnosed on the basis of a history of schistosomiasis and linear calcification on liver CT or linear strong echo on liver ultrasound [14]. MHE was diagnosed by reviewing patient's electronic medical record basing on at least one of following abnormal results: (1) traditional neuropsychological test (NCTs, number connection tests A and B, and DST digit symbol test); (2) new neuropsychological tests (posture control and stability test and multisensory integration test); (3) critical flicker frequency (CFF) test; (4) electroencephalography (EEG); (5) visual evoked potential (VEP) (6) brainstem auditory evoked potential (BAEP); and (7) fMRI [6].

Finally, a total of 236 CHS patients (mean age, 65 years ± 10) were enrolled in this study. The median time between brain MRI scanning and MHE diagnosis was 52 days (range 0–82 days). The patients were randomly assigned into a training group and a testing group according to the ratio of 7:3. An overview of this study's workflow is shown in Fig 1.

## Clinical laboratory tests

Indicators reflecting liver function as serum alanine aminotransferase (ALT), aspartate aminotransferase (AST), total bilirubin (TB), unconjugated bilirubin (UB), prothrombin time (PT), albumin, plasma ammonia and platelet count were recorded.

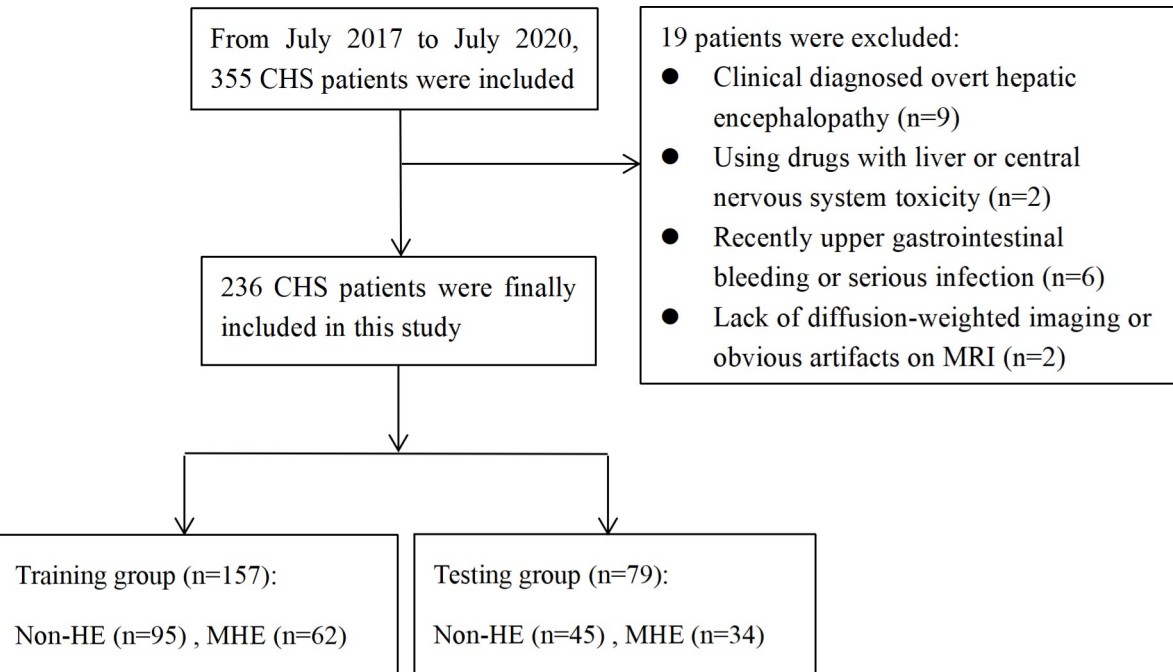

**Fig 1. The workflow of this study.** (CHS: chronic hepatic schistosomiasis; HE: hepatic encephalopathy; MHE: minimal hepatic encephalopathy; DWI: diffusion-weighted imaging; MRI: magnetic resonance imaging).

## MRI acquisition

All brain MRI was performed on a 3.0-Tesla scanner (Verio, Siemens, Erlangen, Germany) with axial T1-weighted imaging (T1WI), T2-weighted imaging (T2WI) and diffusion-weighted imaging (DWI). The details of MRI protocol are presented in S1 Table and Fig 2.

## MRI segmentation and radiomics features extraction

The radiomics data processings were performed referring to a previous work [11]. Briefly, brain MRI from each CHS patient was imported into MITK (http://mitk.org/wiki/MITK), and T2WI and DWI were subsequently aligned to T1WI. Red nuclei, substantia nigra, globus pallidum and subthalamus (region of interest, ROI) were manually drawn on each slice of DWI referring to T2WI and T1WI by a radiologist (Reader 1, with 10 years' experience in brain MRI) blinded to the patients' clinical information. One month later, 50 out of the patients were randomly chosen and the same manual drawings were repeated by Reader 1 and by another radiologist (Reader 2, with 3 years' experience in brain MRI). ROIs were generated a volume region of interest (VOI). Intraclass and interclass correlation coefficients (ICCs) were calculated.

The MR imaging registration and extraction of radiomics features from DWI were performed by using python (Version 3.8.2; https://www.python.org/) "Nipype" package and "pyradiomics" package, respectively. All radiomics features extraction followed the IBSI recommendation (https://arxiv.org/abs/1612.07003).

## Feature selection in training group

Radiomics features with ICC < 0.75 were identified as unstable features. The features with a high correlation with another feature (Pearson's correlation coefficients > 0.9) were identified as redundant features. Unstable features and redundant features (with the largest mean absolute correlation) were removed.

A binary least absolute shrinkage and selection operator (LASSO) logistic regression analysis was performed to select the radiomics features. The selected radiomics features were defined as radiomics signature. Radiomics score (radscore) for each patient was calculated using a linear combination of radiomics signature (S1 Fig). Multivariate binary logistic regression analysis was performed to select clinical laboratory tests (clinical predictors) for predicting MHE in the training group.

## Radiomics nomogram building, testing, discrimination and calibration

The radiomics nomogram for discriminating MHE from non-MHE was developed by combining radiomics signature with selected clinical predictors using multivariable logistic regression. A heatmap was used to analyze the correlation between the radiomics features and the selected clinical predictors in the training group. The radiomics nomogram was validated in the testing group. The area under the curve (AUC) of receiver operator characteristic (ROC) was used to evaluate the discrimination performance of the radiomics nomogram in the training and testing groups. Calibration curve was used to assess the goodness of fit of the radiomics nomogram in the training and testing groups.

Clinical decision curve (CDC) analysis was performed to determine the radiomics nomogram's clinical usefulness and to quantify the net benefits at the threshold probabilities.

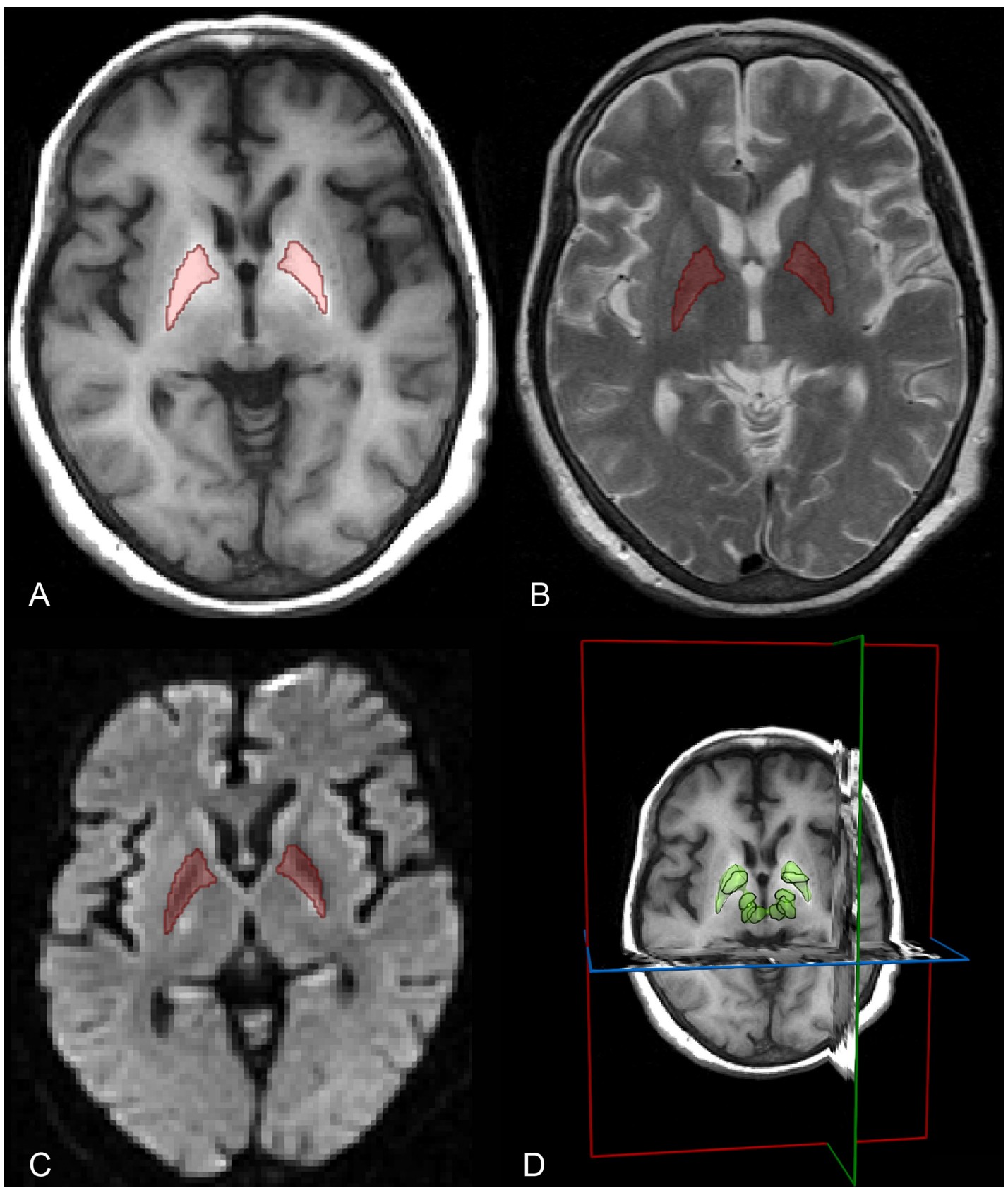

**Fig 2. MR images of a 68-year-old CHS patients with MHE with ROIs.** (A) Axial T1WI marked with ROI drew on bilateral globus pallidum referring to axial T2WI (B) and axial DWI (b = 800 sec/mm$^2$) (C). (D) VOI generated from ROIs of brainstem reticular system (including red nuclei, substantia nigra, globus pallidum and subthalamus). DWI, diffusion-weighted imaging; T1WI, T1-weighted imaging; T2WI, T2-weighted imaging; ROI, region of interest; VOI, volume region of interest.

## Statistical analysis

All statistical analyses were performed in R (Version 4.0.2; http://www.r-project.org/). Student t-test was used for comparing radscore and clinical predictors between non-HE and MHE patients after normality test. Pearson's Chi-square test was used for comparing gender composition between non-HE and MHE patients. Pearson's correlation was used to analyze the correlation between radiomics signature and the clinical predictors. The "caret" package was used for redundant features elimination; the "irr" package was used for ICC calculation; the "glmnet" package was used for binary LASSO logistic regression, linear regression, and multivariate binary logistic regression in selecting radiomics and clinical predictors; the "rms" package was used for nomogram and calibration curve plotting; the "pROC" package was used for AUC calculation; the "dca.R" package was used for decision curve analysis. A P < 0.05 indicated a statistically significant difference.

## Results

### Clinical characteristics of CHS patients

Clinical characteristics of CHS patients in the training and testing groups are summarized in Table 1. The 236 CHS patients included 140 non-HE patients and 96 MHE patients. The mean ages were 66±9 and 65±9 years, respectively (P = 0.471). Eight patients developed into OHE within 1 month after brain MRI scanning without obvious inducing factors. Thirty-two patients were diagnosed from non-HE to MHE in one month follow-up by NCTs A and B, and DST, but none were diagnosed from MHE to non-HE.

No significant differences of AST, ALT, TB, UB were shown between non-HE and MHE patients both in the training and testing groups. Increased plasma ammonia, radscore, and decreased albumin and platelet count were shown in MHE patients than in non-HE patient both in the training and testing groups. Increased PT was shown in MHE patients in the testing group but not in the training group. Multivariate binary logistic regression analysis showed

**Table 1. Comparison of clinical features and radscore between non-MHE and MHE in CHS patients.**

| Features | Training group | | | Testing group | | |
|---|---|---|---|---|---|---|
| | non-MHE n = 98 | MHE n = 42 | P | non-MHE n = 67 | MHE n = 30 | P |
| Age | 65±9 | 64±8 | 0.571 | 67±9 | 66±10 | 0.726 |
| Sex (F/M) | 37/61 | 16/26 | 0.975 | 26/41 | 11/18 | 0.933 |
| ALT | 29.5±14.5 | 30.4±15.9 | 0.697 | 26.5±16.0 | 28.3±16.6 | 0.620 |
| AST | 58.3±11.0 | 61.9±11.1 | 0.054 | 59.8±10.5 | 61.1±9.9 | 0.594 |
| TB | 23.3±9.1 | 22.8±11.8 | 0.779 | 23.3±11.0 | 26.2±12.5 | 0.266 |
| UB | 9.6±3.8 | 9.2±3.8 | 0.498 | 10.4±3.7 | 9.9±4.5 | 0.622 |
| PT | 11±1 | 12±1 | 0.157 | 11±1 | 12±2 | 0.029 |
| Plasma ammonia | 27.4±12.2 | 33.0±16.9 | 0.017 | 29.8±13.1 | 38.8±17.7 | 0.012 |
| Seralbumin | 47.6±18.3 | 40.8±9.4 | 0.003 | 51.6±19.7 | 41.25±9.5 | 0.003 |
| Platelet count | 210±43 | 176±34 | < 0.001 | 209±42 | 176±33 | 0.001 |
| Radscore | 1.51±0.21 | 1.62±0.26 | 0.005 | 1.58±0.25 | 1.65±0.32 | 0.029 |

Aerum alanine aminotransferase (ALT), aspartate aminotransferase (AST), total bilirubin (TB) prothrombin time (PT), unconjugated bilirubin (UB)

that albumin, plasma ammonia and platelet count were clinical predictors for MHE (P < 0.05) (S2 Table).

### Radiomics feature selection

A total of 107 radiomics features were extracted from DWI of each patient. The 91 (85%) features with either interobserver or intraobserver ICC < 0.75 or Pearson's correlation coefficient > 0.9 were removed. After LASSO selection, 6 features were finally remained (defined as radiomics signature) for differentiating between non-HE and MHE (Fig 3A–3C). The heatmap showed the correlation of clinical predictors and radiomics signature in MHE (Fig 3D).

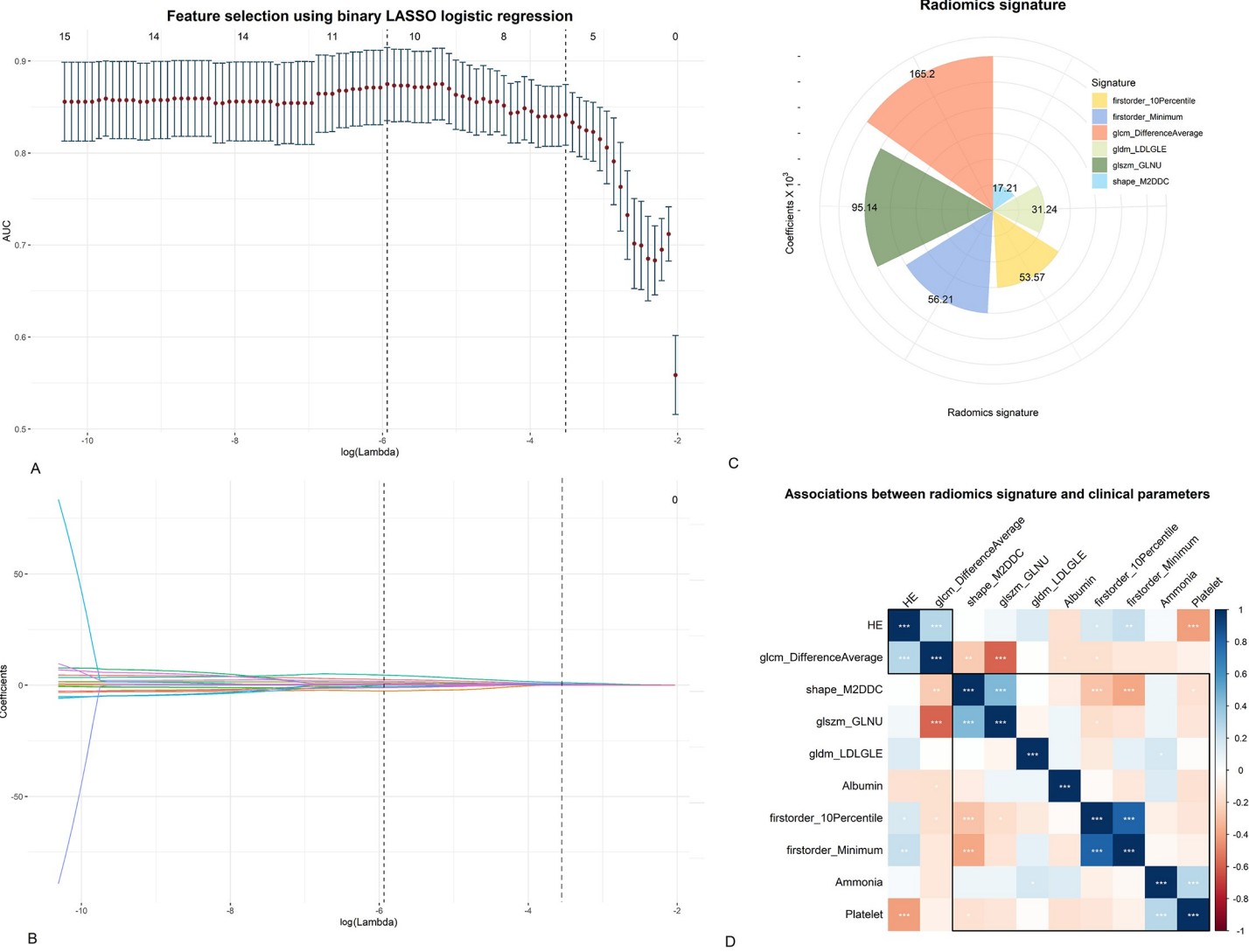

**Fig 3. Process of feature selection for MHE in CHS patients.** The optimal penalty parameter, log (Lambda) is selected at the largest value of log (Lambda) where the error is within one standard error of the minimum criteria, where 6 nonzero coefficients (radiomics signature) have the highest AUC for predicting MHE. (A) Radiomics features are selected by binary LASSO logistic regression. The AUC of MHE is plotted versus log (Lambda). (B) A coefficient profile plot of 6 radiomics features is produced against the log (Lambda). (C) The selected features with their coefficients obtained from the LASSO analysis. (D) A heatmap shows the correlations (by Pearson's correlation) between radiomics features and clinical predictors for MHE. (AUC, area under curve; CHS, chronic hepatic schistosomiasis; LDLGLE, LargeDependenceLowGrayLevelEmphasis; LASSO, least absolute shrinkage and selection operator; M2DDC, Maximum2DDiameterColumn; GLNU, GrayLevelNonUniformity).

## Radiomics nomogram development and performance assessment

A radiomics nomogram by combining the radiomics signature and the clinical predictors (with the lowest Akaike Information Criterion [AIC] score) was developed for predicting MHE (Fig 4A). The calibration curves showed good discrimination performances of the nomogram both in the training and testing groups (Fig 4B and 4C).

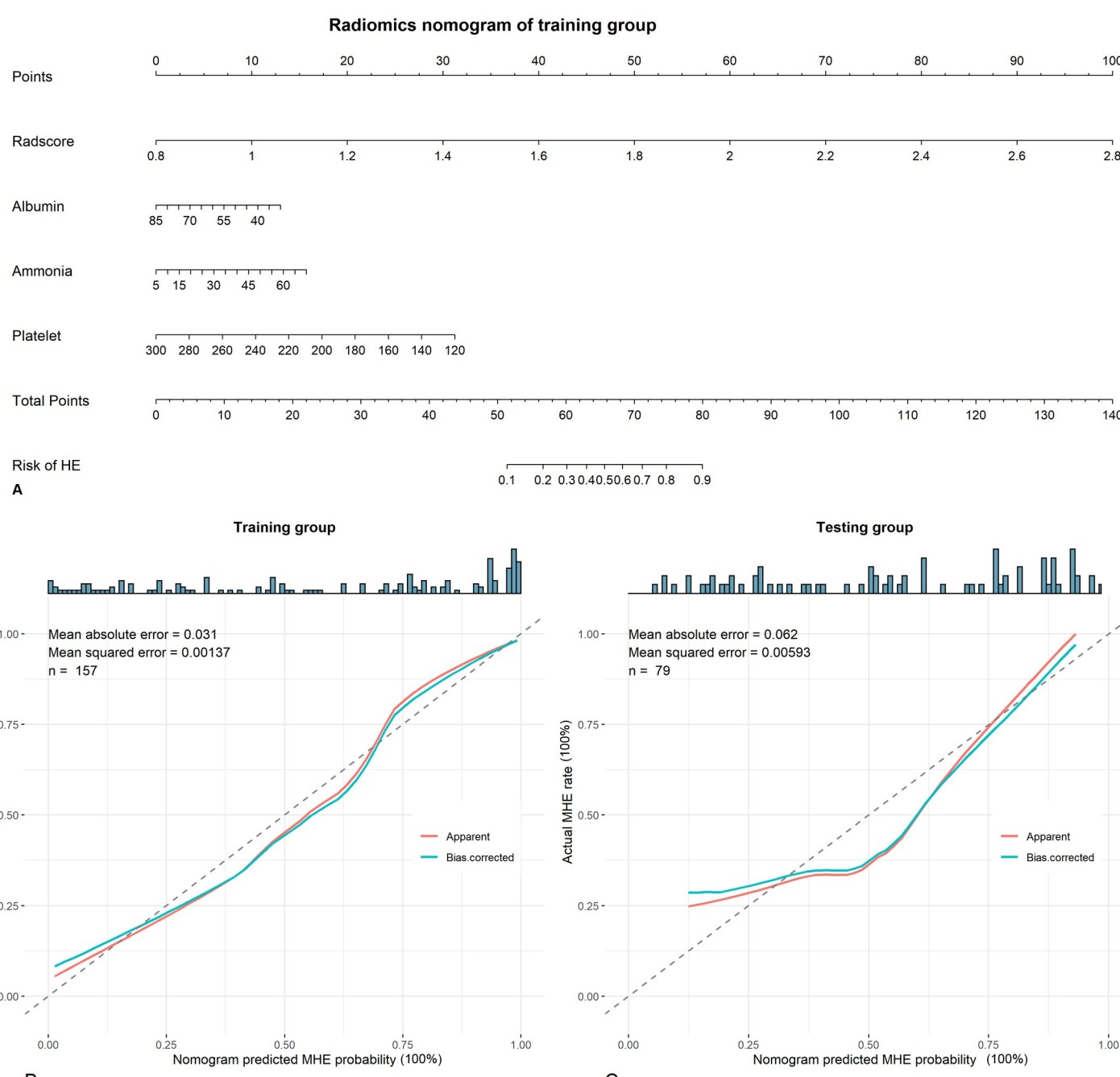

**Fig 4. Radiomics nomogram.** (A) The radiomics nomogram is developed by integrating radscore with seralbumin, plasma ammonia and platelet count in the training group. Calibration curves show goodness of fit both in the training group (B) and the testing group (C).

The sensitivity, specificity, negative predictive value, positive predictive value and AUC of ROC of the radiomics nomogram were 90.3%, 86.3%, 81.2%, 93.2%, 0.89 (95% CI: 0.84–0.94) in the training group; and were 85.3%, 75.6%, 72.5%, 87.2%, 0.83 (95% CI: 0.74–0.92) in the testing group.

### Clinical usefulness of the radiomics nomogram

CDC analyses of the radiomics nomogram for CHS patients with MHE in the training and testing groups are presented in Fig 5. The results showed that the radiomics nomogram for predicting MHE added net benefit both in the training and testing groups (Fig 5).

## Discussion

This retrospective study revealed that the radiomics nomogram by combining radiomics features extracted from brain DWI and clinical predictors (Seralbumin, plasma ammonia and blood platelet) could provide useful information for predicting MHE in CHS patients.

CHS is a important type of chronic liver diseases secondary to cirrhosis from schistosomiasis [15,16]. In the 1950s, about 11.6 million people were infected by schistosomiasis japonicum, while 100 million people were at risk in China [17]. CHS is characterized by progressive liver fibrosis, portal hypertension and portal-systemic shunting [18]. However, no or mild biochemical evidences of liver dysfunction are showed in compensatory stage CHS patients [19,20]. MHE causing by CHS remains neglected until the patients develop into OHE, upper gastrointestinal bleeding, hepatorenal syndrome or hepatopulmonary syndrome in the late stage of CHS [21,22].

Clinical evaluation and prediction of MHE in CHS patients remains a great challenge in routine physical screening. The problems that need to be resolved include the use of neuroimaging, serum biomarkers and a combination of clinical manifestations and neuropsychological testing methods [6]. This study showed that the radiomics nomogram combining DWI radiomics features and clinical predictors was useful in predicting MHE in CHS patients.

In this study, low seralbumin, low platelet count and high plasma ammonia were found to be the clinical predictors of MHE in CHS patient. Liver biochemical indicators, such as bilirubin, AST, ALT, PT, and platelet count have been routinely tested for suspected HE [6]. Elevated blood ammonia is valuable for HE diagnosis. Studies have shown that HE patients often have elevated blood ammonia, however, the degree of elevation does not completely correlate with the severity of HE [23]. It has been reported that liver dysfunction and elevated blood ammonia are not common in CHS patients [23]. Low seralbumin, low platelet count and high plasma ammonia suggest the development of an advanced liver disease due to chronic hepatic injury.

DWI is a method calculating a diffusivity value to quantitatively assess the water molecule movement in tissue on the basis of differences in the mobility of protons (primarily associated with water) [24]. Previous study showed that increased mean diffusivity on diffusion tensor imaging (DTI), increase of cerebral blood volume and flow on arterial spin labeling (ASL), and increase oxygen metabolism rate on functional MRI (fMRI) in CHS patients with MHE [24]. Mean ADC values were significantly increased in caudate, putamen, and pallidus nuclei except thalamus in patients with cirrhosis, which was reported be useful in monitoring patients with HE [25]. Our results of radiomics showed that the futures extracted from DWI were useful in differentiating MHE from non-HE in CHS patients.

The radiomics converts the medical images into high-dimensional data by high-throughput extraction of quantitative features. Followed by subsequent data analysis, the radiomics has been used for diagnosis, differential diagnosis, prediction of treatment efficiency and

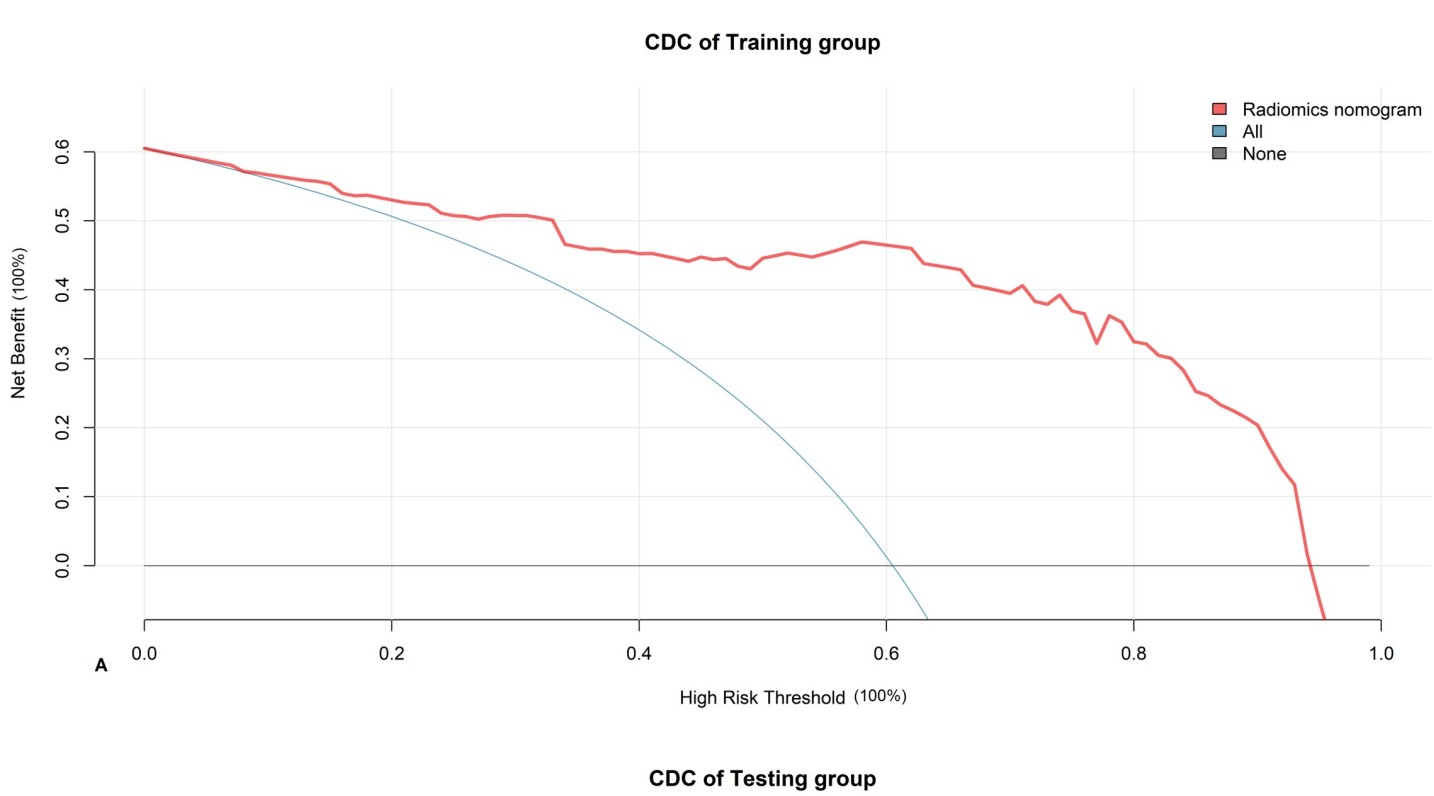

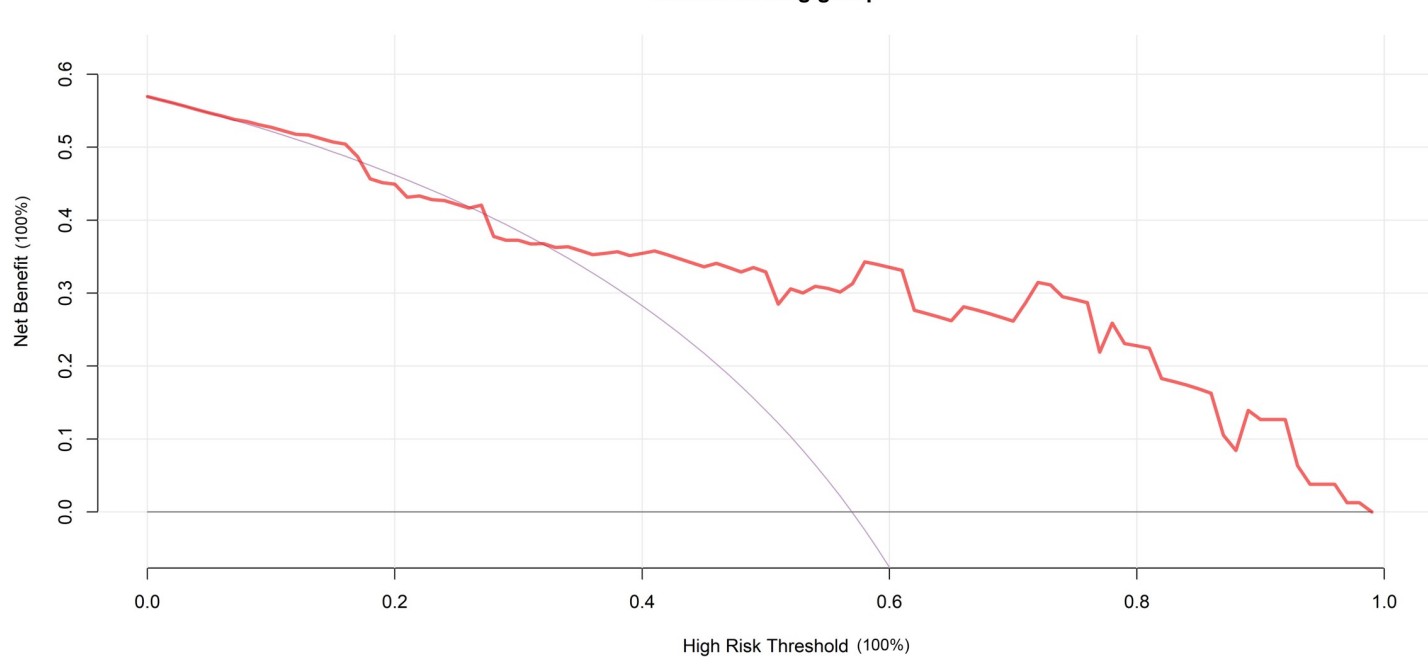

**Fig 5. Clinical decision curve (CDC) analyses.** CDC shows that the radiomics nomogram adds net benefit for predicting MHE both in the training group (A) and testing group (B) than treat all the CHS patients as MHE (blue line) or as non-MHE (black line).

prognosis evaluation [26]. Previously, a radiomics model of liver CT was used to predict the risk of HE in cirrhotics [27]. Nomogram is a diagnostic model that combines imaging and clinical information to determine an individualized prediction or treatment decision [28]. In this study, a radiomics nomogram was developed by integrating the radiomics signature with

clinical risk factors of MHE in CHS patients. The radiomics nomogram was confirmed having the ability to generate a personalized probability to predict MHE in CHS patients. In addition, we applied a CDC analysis, which offers insight into clinical consequences on the basis of threshold probability, and the net benefit of radiomics nomogram in predicting MHE was calculated. The net benefit is defined as the proportion of true positives minus the proportion of false positives, weighted by the relative harm of false-positive and false-negative results [29]. Our results indicated good clinical usefulness of radiomics nomogram in assisting MHE in CHS patients.

This study had several limitations. First, selection bias was inevitable because of the retrospective nature of this study. Second, 8 patients developed into OHE within 1 month after brain MRI scanning without obvious inducing factors, no further statistical analysis was performed due to a small sample size in these patients. Third, 32 patients were diagnosed from non-HE to MHE in one month follow up by NCTs A and B, and DST, but none were diagnosed from MHE to non-HE due to multi-examination methods adopted to identify MHE. Thus, a comparison of radiomics nomogram and a specific examination for predicting MHE was not able to be performed. Furthermore, larger sample, multi-center and prospective studies should be carried out for validating the radiomics nomogram to provide reliable evidence for further clinical application.

## Conclusion

This study developed a radiomics nomogram model by combining DWI radiomics features and clinical predictors of MHE in CHS patients. The radiomics nomogram had a good diagnostic performance in predicting MHE in CHS patients.

## Supporting information

**S1 Table. MRI examination's parameters.**
(DOCX)

**S2 Table. Logistic regression analyses results of clinical predictors.**
(DOCX)

**S1 Fig. Distributions of the radscore.** The radscore of each CHS patients with MHE (rad bar) in the training group (A) and testing group (B). CHS, chronic hepatic schistosomiasis; MHE, minimal hepatic encephalopathy.
(TIF)

## Author Contributions

**Conceptualization:** Ying Li, Jin Wei Qiang.

**Formal analysis:** Ying Li, Xin Li.

**Funding acquisition:** Ying Li, Jin Wei Qiang.

**Investigation:** Ying Li, Shuai Ju, Xin Li.

**Methodology:** Ying Li, Shuai Ju.

**Supervision:** Yan Li Zhou, Jin Wei Qiang.

**Writing – original draft:** Ying Li.

**Writing – review & editing:** Ying Li, Jin Wei Qiang.

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
