## [Decision Letter · Decision Letter 0]

6 Jul 2021

Dear Mr. Qiang,

Thank you very much for submitting your manuscript "Prediction of minimal hepatic encephalopathy by using an radiomics nomogram in chronic hepatic schistosomiasis patients" for consideration at PLOS Neglected Tropical Diseases. As with all papers reviewed by the journal, your manuscript was reviewed by members of the editorial board and by several independent reviewers. In light of the reviews (below this email), we would like to invite the resubmission of a significantly-revised version that takes into account the reviewers' comments. 

We cannot make any decision about publication until we have seen the revised manuscript and your response to the reviewers' comments. Your revised manuscript is also likely to be sent to reviewers for further evaluation.

Sincerely,

Matty Knight, Ph.D

Associate Editor

Michael Hsieh

Deputy Editor

Reviewer's Responses to Questions

**Key Review Criteria Required for Acceptance?**

**Methods**

-Are the objectives of the study clearly articulated with a clear testable hypothesis stated?

-Is the study design appropriate to address the stated objectives?

-Is the population clearly described and appropriate for the hypothesis being tested?

-Is the sample size sufficient to ensure adequate power to address the hypothesis being tested?

-Were correct statistical analysis used to support conclusions?

-Are there concerns about ethical or regulatory requirements being met?

Reviewer #1: The authors appeared to perform their analysis carefully. However, the evaluation does not appear to be thorough. The current work evaluated only the predicting performance of the combined features. It is difficult to state that integrating two feature sets is useful without proper comparison or analysis. The authors should have evaluated two feature sets (radiomic features and clinical risk factors) separately first. Then, the combined radiomic and clinical features could be compared to see any improvement in predicting performance. Ideally, clinical features vs. radiomic features vs. combined features vs. combined features with selection (reduced combined features that are reported in this manuscript) should have been analyzed to draw the authors' conclusion. 

The analysis needs to include more performance metrics, even if they may appear to be redundant. Machine-learning studies provide multiple performance metrics in general. In addition, conducting cross-validation would support the authors' methodology. 

Lines 108-110: The volume of interest is enough (or 3-D region of interest). This reviewer noticed a bit confusing use of the acronym of ICC. ICC is used to abbreviate the intraclass correlation coefficient. Interclass correlation coefficients are usually represented by their common names, e.g., Pearson's correlation coefficient. This reviewer recommends use ICC only for intraclass correlation coefficients. The current statement here appears to be ICC used for both intraclass and interclass correlation coefficients.

Line 116: Please, clarify 'unstable' features. 

Line 118: Please, clarify the term 'redundant.' Does this mean that a redundant feature strongly correlates to many other features simultaneously (for example., collinearity leading to variance inflation due to correlated independent predictors in a multiple regression model)?

Lines 121-123: This is a potentially very important approach unless it has been published somewhere else. Its description appears to be similar to the concept of the latent variable from PCA (principal component analysis). The authors must spell out the "linear combination" approach used to created radscore. It is hidden in the current manuscript. If it has been published somewhere else, please provide its reference.

Reviewer #2: Instead of removing the "redundant features" a dimension reduction algorith could be an option.

Why clinical features are handled by a separate model?

What is the contribution of the radiomics features to the model?

Which softare has been used for the statistical analysis?

Are the authors applying the model developed in the training set to the testing set? From the results section it appears that the same features, not the same parameters have been applied. The authors should predict what happen in the testing set using the model parameters from the training set.

**Results**

-Does the analysis presented match the analysis plan?

-Are the results clearly and completely presented?

-Are the figures (Tables, Images) of sufficient quality for clarity?

Reviewer #1: Results section needs major revision.

The current result section is overly brief and needs more elaboration of tables/figures with quantitative information. It should not include descriptions appropriate for the methods section.

Reviewer #2: The results need to be revised according to the new methods

**Conclusions**

-Are the conclusions supported by the data presented?

-Are the limitations of analysis clearly described?

-Do the authors discuss how these data can be helpful to advance our understanding of the topic under study?

-Is public health relevance addressed?

Reviewer #1: Current conclusion based on the presented methods and results did not convince this reviewer. 

After having compared the two feature sets separately and combined, a more detailed discussion can be elaborated on the value and implications of integrating radiomics and clinical risk factors.

Reviewer #2: Some concerns on diagnostic nomograms developed with small sample size.

**Editorial and Data Presentation Modifications?**

Reviewer #1: (No Response)

Reviewer #2: (No Response)

**Summary and General Comments**

Reviewer #1: This manuscript is well organized and delivers an interesting application of a machine-learning methodology in a clinical study setting. The primary objective of this study is to evaluate a data-driven method developed to predict the minimal hepatic encephalopathy (MHE) in chronic hepatic schistosomiasis (CHS) patients. 

Although the topic of the study will benefit clinicians and researchers in this field, the current form of the manuscript raised a few concerns. As such, this reviewer would like to encourage the authors to reflect comments and suggestions provided by this reviewer on the next iteration of the manuscript.

Reviewer #2: (No Response)

PLOS authors have the option to publish the peer review history of their article (what does this mean?). If published, this will include your full peer review and any attached files.

Reviewer #1: No

Reviewer #2: No
---

## [Decision Letter · Decision Letter 1]

23 Sep 2021

Dear Dr. Qiang,

We are pleased to inform you that your manuscript 'Prediction of minimal hepatic encephalopathy by using an radiomics nomogram in chronic hepatic schistosomiasis patients' has been provisionally accepted for publication in PLOS Neglected Tropical Diseases.

Before your manuscript can be formally accepted you will need to make all corrections suggested by reviewer 1 and complete some formatting changes, which you will receive in a follow up email. A member of our team will be in touch with a set of requests.

Best regards,

Matty Knight, Ph.D

Associate Editor

Michael Hsieh

Deputy Editor

This is accepted provided the suggestions made by reviewer 1 are adhered to in improving the manuscript.

Reviewer's Responses to Questions

**Key Review Criteria Required for Acceptance?**

**Methods**

-Are the objectives of the study clearly articulated with a clear testable hypothesis stated?

-Is the study design appropriate to address the stated objectives?

-Is the population clearly described and appropriate for the hypothesis being tested?

-Is the sample size sufficient to ensure adequate power to address the hypothesis being tested?

-Were correct statistical analysis used to support conclusions?

-Are there concerns about ethical or regulatory requirements being met?

Reviewer #1: This reviewer find no issues in the method section.

**Results**

-Does the analysis presented match the analysis plan?

-Are the results clearly and completely presented?

-Are the figures (Tables, Images) of sufficient quality for clarity?

Reviewer #1: In Fig2, the figure legend does not properly describe the figures. A, B, C, and D are not shown in the figure legend. A and B appear to be similar slices. However, DWI does not appear to be a similar slice judging from the shape of the ventricles. In the legend, T1W and T2W are elaborated. However, DWI was not spelled out. The colored ROI was overlaid only on A. It would be useful to overlay the VOI on B and C assuming they are co-registered and in the same space. In D, showing VOI in a semi-transparent background brain image will be helpful.

In Fig 3, either Log or log should be used in a consistent manner. In B, what coefficients? Lambda is shown as a Greek letter in the legend, not in the figure. It has to be shown in a consistent manner throughout the figures, the figure legend, and the manuscript. In D, the types of associations should be specified. If that is not a single type, the legend should show a proper explanation since the associations can be presented and measured in various statistics.

In Fig 4, the authors should provide proper units for a few variables. In B and C, the vertical axes show some rates, which should have some units.

In Fig 5, there are no apparent units for both variables. In this case, the authors can use (units: arbitrary), which is very common for unit-less variables (hence, the axes).

**Conclusions**

-Are the conclusions supported by the data presented?

-Are the limitations of analysis clearly described?

-Do the authors discuss how these data can be helpful to advance our understanding of the topic under study?

-Is public health relevance addressed?

Reviewer #1: In the discussion section, either vs. or VS. should be used consistently.

As in 'training set and test set,' training group and test group would be an appropriate naming convention in the machine learning field.

**Editorial and Data Presentation Modifications?**

Reviewer #1: By incorporating minor comments into this manuscript, the current draft could be enhanced for the readership.

**Summary and General Comments**

Reviewer #1: The authors have significantly improved the clarity of the text and the presentation of the results in this current version of the manuscript. After reading the current manuscript several times, this reviewer has a few minor, not major, comments.

PLOS authors have the option to publish the peer review history of their article (what does this mean?). If published, this will include your full peer review and any attached files.

Reviewer #1: No

---

## [Editor Report · Acceptance letter]

13 Oct 2021

Dear Mr. Qiang,

We are delighted to inform you that your manuscript, "Prediction of minimal hepatic encephalopathy by using an radiomics nomogram in chronic hepatic schistosomiasis patients," has been formally accepted for publication in PLOS Neglected Tropical Diseases.

Best regards,

Shaden Kamhawi

co-Editor-in-Chief

Paul Brindley

co-Editor-in-Chief
